# Rapid diagnosis of periodontitis, a feasibility study using MALDI-TOF mass spectrometry

**Angéline Antezack[1,2], Hervé Chaudet[2], Hervé Tissot-Dupont[2], Philippe Brouqui[2], Virginie Monnet-Corti[1,2] ***

**1** Department of Periodontology, Service of Odontology, AP-HM, UFR of Odontology, Aix-Marseille University, Marseille, France, **2** AP-HM, IHU-Méditerranée Infection, Institut de Recherche pour le Développement, Institut Hospitalo-Universitaire Méditerranée Infection, MEPHI, Aix Marseille University, Marseille, France

* virginie.corti@univ-amu.fr

**Data Availability Statement:** All relevant data are within the manuscript and its Supporting Information files.

## Abstract

### Aim

The aim of the present study was to assess the feasibility and diagnostic contribution of protein profiling using MALDI-TOF mass spectrometry applied to saliva, gingival crevicular fluid (GCF) and dental plaque from periodontitis and healthy subjects. We hypothesized that rapid routine and blinded MALDI-TOF analysis could accurately classify these three types of samples according to periodontal state.

### Materials and methods

Unstimulated saliva, GCF and dental plaque, collected from periodontitis subjects and healthy controls, were analyzed by MALDI-TOF MS. Based on the differentially expressed peaks between the two groups, diagnostic decision trees were built for each sample.

### Results

Among 141 patients (67 periodontitis and 74 healthy controls), the decision trees diagnosed periodontitis with a sensitivity = 70.3% (± 0.211) and a specificity = 77.8% (± 0.165) for saliva, a sensitivity = 79.6% (± 0.188) and a specificity = 75.7% (± 0.195) for GCF, and a sensitivity = 72.1% (± 0.202) and a specificity = 72.2% (± 0.195) for dental plaque. The sensitivity and specificity of the tests were improved to 100% (CI 95% = [0.91;1]) and 100% (CI 95% = [0.92;1]), respectively, when two samples were tested.

### Conclusion

We developed, for the first time, diagnostic tests based on protein profiles of saliva, GCF and dental plaque between periodontitis patients and healthy subjects. When at least 2 of these samples were tested, the best results were obtained.

**Funding:** This work was supported by the French Government under the 'Investissements d'avenir' (Investments for the Future) program managed by the AgenceNationale de la Recherche (ANR, fr: National Agency for Research) (reference: Méditerranée Infection 10-IAHU-03).

**Competing interests:** The authors have declared that no competing interests exist.

## Introduction

Periodontitis is a chronic multifactorial inflammatory disease associated with dysbiotic plaque biofilms that results in the progressive destruction of the supporting structures of the teeth [1]. The 2009–2012 NHANES estimated that 46% of adults aged 30 years or older in the United States had periodontitis with 8.9% having two or more interproximal sites with $\geq 6$ mm clinical attachment loss (not on the same tooth) and one or more interproximal site(s) with $\geq 5$ mm periodontal probing depth [2]. Due to its high prevalence and its consequences on quality of life, oral health and associations with systemic diseases, periodontitis is an important public health problem [3,4]. The diagnosis of periodontitis is currently based on clinical measurements of probing pocket depth (PD), bleeding on probing (BOP), plaque index, and clinical attachment level and is associated with a radiographic examination [1]. The major drawbacks of this clinical diagnosis are that it is time-consuming and that it requires professional skills and training, making it difficult to use for large-scale screening. Furthermore, because at the beginning of periodontal disease the patient does not experience any symptoms, he will not consult and will not be diagnosed until an advanced stage of periodontitis is reached. These assessments highlight the necessity to have a simple minimally invasive screening and diagnosis tool for periodontal health and periodontal disease at any stage.

Recently, the development of diagnostic tests based in either oral fluid or blood analysis has seen considerable growth for population screening in many medical disciplines [5–7]. Furthermore, the use of oral fluid as an alternative to venous blood has been intensively explored in various fields of research such as therapeutic drug monitoring or diseases biomarker discovery purpose [8,9]. Compared to blood and its derivatives, saliva carries many advantages including non-invasiveness, no need for highly trained personnel, safer to handle, ease to collect, ship and store [10]. Recently, Campanella et al. have found that analysis of volatile salivary metabolites could be promising for the indirect assessment of gut microbiota [11].

In the periodontal field, numerous studies have attempted to investigate potential changes in saliva, gingival crevicular fluid (GCF) and dental plaque that could be correlated to periodontitis and be used to develop new alternative diagnostic method [12–14]. However, to date, no unique biomarker has been identified as sensitive and specific enough to be used for diagnosing periodontitis [15,16]. Biological tests are still dramatically missing in the periodontal diagnosis field while they already exist or will exist tomorrow for other diseases [17,18]. Faced with this observation, the evaluation and comparison of global protein profiles seem more relevant than the search for and identification of biomarkers in the periodontitis diagnosis. The existence of specific protein profiles could represent the signature of a periodontal disease.

Matrix-assisted laser desorption ionization time-of-flight (MALDI-TOF) mass spectrometry (MS) is a rapid and precise proteome profiling method that generates a characteristic spectrum for analytes in the sample. Profiling is a conventional approach that is widely used in bacterial identification by comparing spectra of unknown bacteria to databases that contain spectra of reference bacteria [19–20]. In this non-quantitative technique, intact cells can directly be mixed with MALDI matrix [20]. La Scola & Raoult have shown that this method was efficient for direct routine identification of bacterial isolates in blood culture and may surpass the conventional diagnostic methods in speed and accuracy [21]. Beyond the realms of microbial world, recent studies demonstrated that MALDI-TOF MS technique can be used to identify fleas [22], ticks [23] and mosquitoes [24] and also appears promising in the rapid diagnosis of cancerous lung nodule [25].

In the periodontology field, Ngo et al. were the first to use mass spectrometric techniques to identify 33 peptides and 66 proteins in GCF from inflammatory sites in periodontal

maintenance subjects [26]. In another study, the same authors found that peptide profiles of GCF from periodontal maintenance subjects could be used to predict sites with attachment loss [27]. Zhang et al. compared the protein profiles of orthodontics patients with and without periodontitis using MALDI-TOF MS and found significantly different intensities of nine peaks, seven of which were higher in healthy subjects [28]. Chaiyarit et al. analyzed salivary protein profiles in oral cancer, oral lichen planus, and chronic periodontitis and found that mass signals at 5,835.73 and 9,801.83 Da were significantly decreased in periodontitis patients compared to the two others oral diseases [29]. More recently, in a small number of patients, changes in protein profiles in chronic periodontitis were reported in saliva, GCF and serum [30].

The few studies identifying protein profiles changes were case-control studies, which described differences between periodontitis and healthy subjects, but there are no data available from blinded experiments. The aim of the present study was to assess the feasibility and diagnostic contribution of protein profiling using MALDI-TOF mass spectrometry applied to saliva, GCF and dental plaque from periodontitis and healthy subjects. We hypothesized that rapid routine and blinded MALDI-TOF analysis could accurately classify these three types of samples according to periodontal state.

## Materials and methods

This study was approved by the Institutional Review Board (IRB) and Independent Ethics Committee (IEC) of Institut Hospitalo-Universitaire (IHU), Microbes Evolution Phylogeny and Infections—(MEPHI) under N° 2019–008. Informed written consent was obtained from each patient.

### Study population

The subject population was recruited from January 2019 until June 2019. Periodontitis subjects were recruited among patients referred to the periodontal department of the Pavillon Odonto-logique de la Timone, Marseille. Control subjects were recruited among young adults. A medical questionnaire (epidemiological data) including the main general risk factors for periodontal diseases was completed (S1 Table) and a periodontal examination was performed, including an intra-oral examination and full-mouth periodontal probing. A periodontal diagnosis was made, and subjects were divided into two groups: the periodontitis group, in which the periodontitis has been classified into stages and grades according the Chicago classification [1] and the healthy periodontium (control) group.

According to the Chicago classification [1], the inclusion criteria for:

- the periodontitis group were interdental clinical attachment loss (CAL) detectable at $\geq 2$ non-adjacent teeth or buccal or oral CAL $\geq 3$ mm with pocketing $\geq 3$ mm detectable at $\geq 2$ teeth,

- the healthy periodontium control group were bleeding score on probing $< 10\%$, pocket depth $< 3$ mm and no clinical attachment loss.

All the patients were selected based upon periodontal status, regardless of the other criteria (e.g., systemic disease or disease/infection that may affect the periodontal health status, use of antibiotics or immunosuppressant medication within 3 months, current or former smokers), in order to reduce selection bias.

Non-inclusion criteria for both groups were history of periodontal therapy within the previous 6 months, pregnant/lactating women and orthodontic patients.

## Sample collections

**Saliva.** All subjects were asked to not eat, drink or brush their teeth one hour before sample collection. Each subject was asked to expectorate whole saliva into a 50-mL centrifuge tube until a minimum saliva volume of 2 mL was collected.

**GCF and subgingival dental plaque.** GCF and subgingival dental plaque samples were collected from one site showing PD <3mm without CAL or BOP of one tooth in each quadrant (4 sites from each subject in total) in control group. In periodontitis group, samples were collected from one site with PD ≥ 5mm and CAL ≥ 3mm of one tooth in each quadrant of each subject (4 sites from each subject in total). For both GCF and subgingival dental plaque collections, all supra-gingival plaque facing the sampling area was removed with a sterile curette to avoid contaminating the samples. The site was then isolated by cotton rolls and gently air dried for 5 s to remove any saliva present. GCF was collected using sterile absorbent paper points (Paper points N˚20, VDW-Zipperer®) carefully positioned into the periodontal pocket from periodontitis patients and into the gingival sulcus from control subjects and left for 30 s. GCF within the periodontal pocket or the sulcus was absorbed by the paper points through capillary action. A total of 4 paper points were obtained from each patient and placed into a 1.5-mL Eppendorf tube containing 100 μL HPLC-grade water (HPLC: high-performance liquid chromatography). Subgingival dental plaque was collected using a sterile curette and placed into a 1.5 mL Eppendorf tube containing 100 μL HPLC-grade water. Samples were immediately stored at 4˚C and analyzed within 24 to 48 hours. Only samples not visually contaminated with blood were selected for the study.

**Mass spectrometry.** A volume of 0.5 μL of each sample was directly spotted six times onto a 96 polished steel MALDI target and then allowed to dry at room temperature. Deposits were then coated with 1 μL of a matrix solution containing α-cyano-4-hydroxycinnamic acid diluted into 500 μL of acetonitrile, 250 μL of 10% trifluoroacetic acid and 250 μL of HPLC-grade water. All manipulations were carried out under class II biological safety cabinets MSC-AdvantageTM (Thermo Fischer Scientific, Villebon sur Yvette, France). After drying for a few minutes at room temperature, the target was introduced into a Microflex LT MALDI-TOF mass spectrometer laser (Bruker Daltonics, Bremen, Germany; external mass spectrometer calibration accuracy ± 300ppm). Each sample generated 6 spectra from 6 deposits. Spectra were recorded in the positive linear mode at a laser frequency of 50 Hz within a mass range of 2–20 kDa. The acceleration voltage was 20 kV, and the extraction delay time was 200 ns. Each spectrum was obtained from 240 laser shots performed in 6 regions of the same spot and then automatically acquired using the AutoXecute acquisition control in FlexControl software 3.0 (Bruker Daltonics). The spectra of the six spots for each sample were imported into the BioTyper-RTC$^{TM}$ version 3.0 software (Bruker Daltonics GmbH). The calibration of the MS was fully automated and performed with a commercial solution (BTS: Bacterial standard test) and the procedure was completely automatic (BiotyperRTC user manual). For each analysis, the MALDI target was simultaneously tested with an inactivated strain of *Escherichia coli* as the positive control (objective score > 2,1) and with matrix solution alone as the negative control (objective score < 1,5). All spectra were controlled using the Flexanalysis® v3.4 software (Bruker Daltonics, Bremen, Germany). Quality criteria of the spectrum for global aspect and intensity were checked : intensity above $10^4$ arbitrary units (AU), horizontal baseline curve and presence of visually identifiable peaks.

**Bioinformatics analysis.** Spectra were analyzed using a homemade R program [31] including the supplementary libraries MALDIQuant [32], seriation [33] and binDA [34].

A first step of spectra pre-processing included noisy spectra discarding using spectrum signal to noise ratio, smoothing (moving average with half window size 8), baseline correction

(Statistics-sensitive Non-linear Iterative Peak-clipping algorithm, 100 iterations), intensity recalibration (total ion current), peak selection (MAD with half window size 8 and signal-noise ratio threshold 3), spectra alignment (quadratic warping function with 0.002 tolerance), averaging of technical replicates in main spectrum profiles (MSP), peak binning, and intensity matrix building, as recommended by Gibb & Strimmer [35].

Differences in the two groups analyzed were assessed on the basis of a discriminant peak identification list using binary predictors. To create the list of discriminant peaks for each sample, we performed a discriminant analysis between groups using the Binary Discriminant Analysis method [34]. Then, we searched for models able to correctly discriminate the two groups, periodontitis and healthy periodontal subjects, from each sample. We generated binary decision trees using Quinlan's C5.0 algorithm, an extension of C4.5 [36]. In a second step, an internal validation was processed with a 10-fold cross-validation in order to verify the classification ability of the generated models. Each group of samples was randomly partitioned into 10 equal-sized subsamples. One subsample was selected as the validation data for testing the diagnostic decision tree, while the other 9 subsamples were used as training data. The process was repeated 10 times with a rotation of the subsamples tested. The performance of the models of each sample was evaluated by sensitivity and specificity. Finally, a principal component analysis was carried out between the top 10 ranking peaks and the epidemiological data from the medical questionnaire (age, gender, current smokers, former smokers, diabetes, cardiovascular disease, hypothyroidism, arthritis, respiratory disease, anti-diabetic medication, antibiotics, anti-inflammatory, antihypertensive, anticoagulants, thyroid hormone thyroxine, hormonal contraception and stress).

## Results

### Characteristics of the subjects in the study

A total of 141 subjects, specifically 39 males and 102 females aged from 20 to 77 years, were enrolled in the study. The periodontitis group included 67 subjects (mean age 50.18 ± 13.85 years, 14 males/53 females). The control group included 74 subjects (mean age 24.50 ± 3.28 years, 25 males/49 females). Patient details are listed in Tables 1 and 2. The same proportions of current smokers were present in our two groups. A large proportion of our periodontitis group was classified into stages III (46.3%) and IV (41.8%). A total of 119 samples of saliva (51 periodontitis; 68 controls), 104 samples of GCF (54 periodontitis; 50 controls) and 110 samples of dental plaque (51 periodontitis; 59 controls) were collected.

**Differentially expressed peptide peaks.** Saliva, GCF and dental plaque samples were analyzed by MALDI-TOF MS, and protein profiles were obtained from each sample in the range of 2–20 kDa.

A total of 217 peaks were detected in saliva, among which 114 were significantly different between the two groups (p < 0.05) (S2 Table). Among the top 10 ranking peaks, 8 (m/z values: 3372 Da, 3443 Da, 3519 Da, 3550 Da, 6352 Da, 6735 Da, 12692 Da, and 13461 Da) had higher levels of intensities in the periodontitis group, and 2 (m/z values: 2620 Da and 7746 Da) had higher levels of intensity in the control group (Fig 1A).

A total of 176 peaks were detected in GCF, among which 110 were significantly different between the two groups (p < 0.05) (S3 Table). Among the top 10 ranking peaks, 9 (m/z values: 3775 Da, 4235 Da, 5296 Da, 5728 Da, 5893 Da, 10586 Da, 11324 Da, 11359 Da and 11447 Da) had higher levels of intensity in the periodontitis group, and only 1 (m/z values: 4944 Da) had a higher level of intensity in the control group (Fig 1B).

A total of 124 peaks were detected in dental plaque, among which 54 were significantly different between the two groups (p < 0,05) (S4 Table). All top 10 ranking peaks (m/z values:

**Table 1. Demographic characteristics contrasting the subjects from the periodontitis group and control group, with the p-value resulting from the between-group comparison of all subjects.**

| | Periodontitis group | | | Control group | | | p value |
|---|---|---|---|---|---|---|---|
| Variable | Male | Female | All | Male | Female | All | |
| Number of patients | 14 | 53 | 67 | 25 | 49 | 74 | |
| Age (years) | 49.14 ± 15.35 | 50.45 ± 13.58 | 50.18 ± 13.85 | 24.12 ± 2.35 | 24.63 ± 3.67 | 24.50 ± 3.28 | <0.0001* |
| Number of current smokers | 4 | 14 | 18 | 8 | 9 | 17 | 0.593 |
| Number of former smokers | 5 | 11 | 16 | 2 | 2 | 4 | 0.0017* |
| Diabetes (HbA1c < 7) | 0 | 4 | 4 | 0 | 0 | 0 | 0.0330* |
| Cardiovascular disease | 2 | 6 | 8 | 1 | 2 | 3 | 0.0812 |
| Hypothyroidism (TSH<4 mUI) | 2 | 8 | 10 | 0 | 1 | 1 | 0.0027* |
| Arthritis | 0 | 2 | 2 | 0 | 1 | 1 | 0.502 |
| Respiratory disease | 2 | 2 | 4 | 1 | 2 | 3 | 0.600 |
| Anti-diabetic medication[†] | 0 | 4 | 4 | 0 | 0 | 0 | 0.0330* |
| Antibiotics[†] | 0 | 2 | 2 | 0 | 3 | 3 | 0.731 |
| Anti-inflammatory[†] | 0 | 2 | 2 | 2 | 9 | 11 | 0.0149* |
| Antihypertensive[†] | 2 | 4 | 6 | 0 | 1 | 1 | 0.037* |
| Anticoagulants[1] | 1 | 5 | 6 | 0 | 1 | 1 | 0.0379* |
| Thyroid hormone thyroxine (T4)[†] | 2 | 6 | 8 | 0 | 41 | 1 | 0.0102* |
| Hormonal Contraception[†] | 0 | 11 | 11 | 0 | 21 | 21 | 0.0904 |
| Stress | 4 | 25 | 29 | 10 | 31 | 41 | 0.150 |

[†]Medication within a month

*Significant difference (p<0.05)

2407 Da, 2627 Da, 2783 Da, 2818 Da, 3038 Da, 3077 Da, 3194 Da, 4065 Da, 4931 Da and 14693 Da) had lower levels of intensity in the periodontitis group (Fig 1C).

## Construction of the informatics decision tree for periodontal diagnosis

On the basis of protein profiles obtained by MALDI-TOF MS, a diagnostic decision tree for periodontitis based on the differentially expressed peaks was made for each type of sample (Fig 2). For saliva, 7 peaks were selected to build a diagnostic decision tree with a sensitivity = 98% and a specificity = 91%. For GCF and dental plaque, 9 peaks were selected to build a diagnostic decision tree with a sensitivity = 96% and 94% and a specificity = 98% and 96%, respectively.

In the blind experiment with a 10-fold cross-validation, diagnostic decision trees were confirmed with a sensitivity = 70.3% (± 0.211) and a specificity = 77.8% (± 0.165) for saliva, a sensitivity = 79.6% (± 0.188) and a specificity = 75.7% (± 0.195) for GCF, and a sensitivity = 72.1% (± 0.202) and a specificity = 72.2% (± 0.195) for dental plaque.

When a concatenation of the 3 decision trees was performed with a minimum of two samples tested, we found a sensitivity = 1 (CI 95% = [0.91;1]), a specificity = 1 (CI 95% = [0.92;1]),

**Table 2. Stages and grades in the periodontitis group according to the Chicago classification [1].** The percentage in each box refers to the relative frequency in relation to the total of 67 periodontitis subjects.

| Stage / Grade | 1 | 2 | 3 | 4 | Total |
|---|---|---|---|---|---|
| A | 0 (0,0%) | 3 (4,5%) | 0 (0,0%) | 0 (0,0%) | 3 (4,5%) |
| B | 0 (0,0%) | 5 (7,5%) | 15 (22,4%) | 6 (8,9%) | 26 (38,8%) |
| C | 0 (0,0%) | 0 (0,0%) | 16 (23,9%) | 22 (32,8%) | 38 (56,7%) |
| Total | 0 (0,0%) | 8 (11,9%) | 31 (46,3%) | 28 (41,8%) | 67 (100%) |

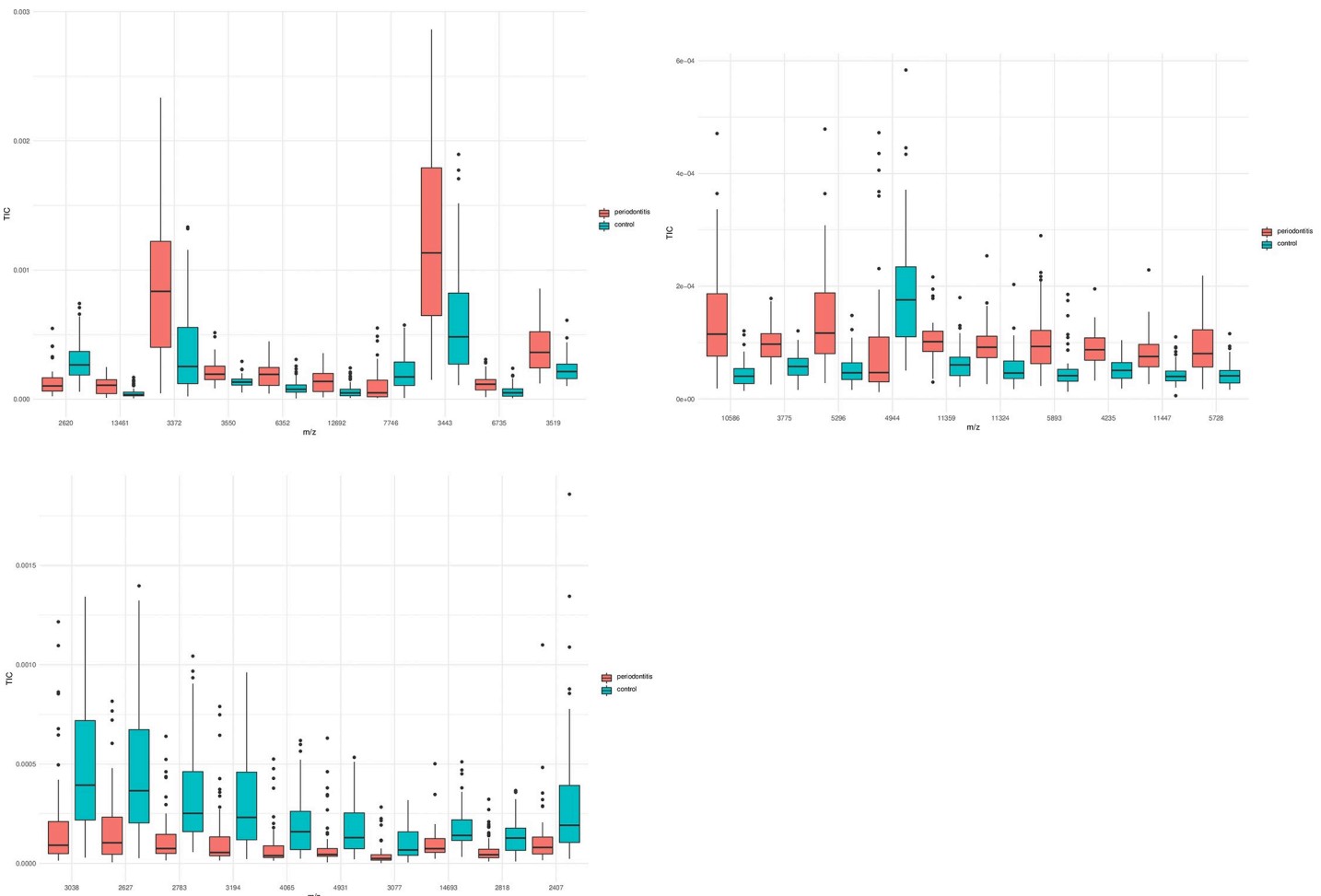

**Fig 1.** Box plots of the 10 top ranking peaks differentially expressed between the periodontitis and the control groups (a. saliva, b. GCF, c. dental plaque). Most of the differentially expressed peaks had higher levels of intensities in the periodontitis group in saliva and GCF, while all 10 top ranking peaks were found to be decreased in dental plaque from the periodontitis group.

a positive predictive value = 1 (CI 95% = [0.91;1]) and a negative predictive value = 1 (CI 95% = [0.92;1]) for our study population.

## Relation between the peaks and the epidemiological data

A principal component analysis was carried out between the 10 top ranking peaks and the epidemiological data from the medical questionnaire (age, gender, current smokers, former smokers, diabetes, cardiovascular disease, hypothyroidism, arthritis, respiratory disease, anti-diabetic medication, antibiotics, anti-inflammatory, antihypertensive, anticoagulants, thyroid hormone thyroxine, hormonal contraception and stress). The 10 top ranking peaks were found independent of the general risk factors for periodontal diseases, which means that these peaks are specific of the periodontal status.

## Discussion

Due to high and constantly increasing prevalence of periodontitis and its consequences on quality of life, oral and general health, a rapid, minimally invasive and large-scale periodontal

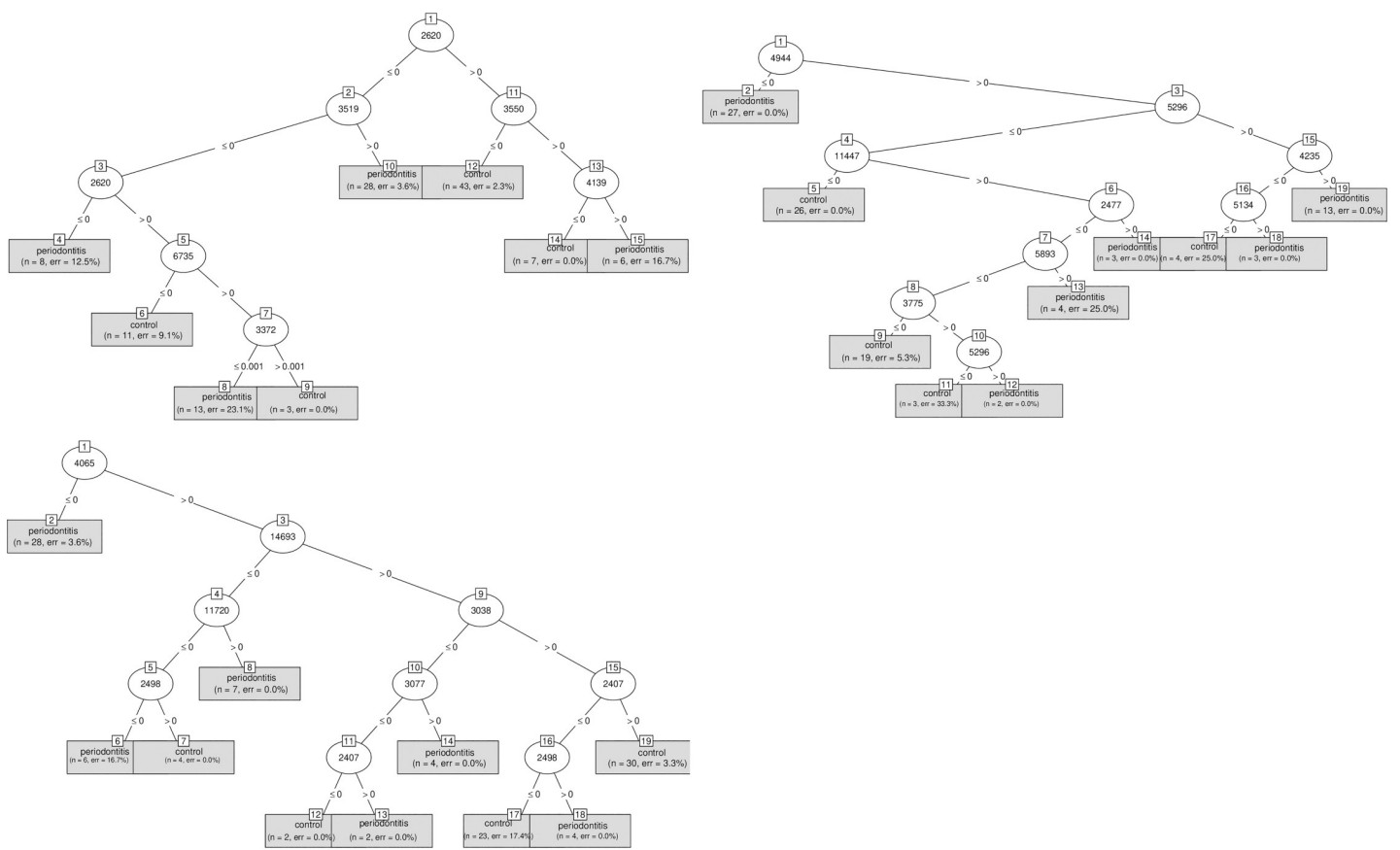

**Fig 2.** Diagnostic decision trees based on differentially expressed peaks between periodontitis and control groups (a. saliva, b. GCF, c. dental plaque).

diagnosis test is of a particular interest, especially in patients with periodontal risk factors. In this study, direct samples analysis of dental plaque, GCF and saliva by a MALDI-TOF mass spectrometer used in routine diagnostic (Microflex LT, Bruker) combined with the use of an algorithm for sample classification resulted in an acceptable performance to correctly classify periodontitis and healthy periodontium subjects. Our study was the first, to our knowledge, to demonstrate that MALDI-TOF MS differentiates periodontitis from healthy periodontium by blind identification of specific patterns in mass signals from protein profiles in saliva, GCF and dental plaque. Previous encouraging results were obtained using MALDI-TOF MS analysis to investigate differences in protein profiles according to the periodontal state, but might be limited by the small sample sizes of the studies [28–30]. In our study, we obtained samples from 141 patients, 67 periodontitis patients and 74 control subjects, and found a large ratio of peaks significantly different between the periodontitis group and the control group in saliva (114 over 217), GCF (110 over 176) and dental plaque (54 over 124). These results reflect that composition of saliva, GCF and dental plaque present important differences in correlation with periodontitis and emphasize that periodontal diagnosis does not depend on a unique biomarker [16]. In comparison, Tang et al. detected 91 salivary peptide peaks, 7 of which were significantly different between the periodontitis and the control group, and 48 in the GCF among 4 were significantly different between the two groups [30]. These different results might be explained by a greater numbers of subjects (119 for saliva and 104 for GCF versus 33) and a larger range of detection (2-20kDa versus 1-10kDa) in our study and also variations in the statistical analysis.

In this study, GCF showed a strong ability to distinguish, in blind experiment, periodontitis patients from control subjects with a sensitivity = 79.6% (± 0.188) and a specificity = 75,7% (± 0,195). This result is not surprising because GCF reflects site-specific periodontal status, while saliva expresses the global status of the oral cavity. Periodontitis has been associated with changes in GCF composition and more than 90 different components have been investigated so far [37–39]. Some of these components have been found to be reduced from healthy subjects to patients with periodontitis, while others have been found to have concentrations positively correlated with periodontal inflammation [40,41].

Saliva profiles allowed periodontitis diagnosis, in blind experiment, with a sensitivity = 70.3% (± 0.211) and a specificity = 77.8% (± 0.165). This result supports the promising interest of saliva as a diagnostic fluid for screening patients with periodontal diseases. Numerous salivary components including locally produced proteins and salivary microbiota have been investigated as potential reflect of periodontal status [42,43]. For example, *Prevotella* has been found to be overabundant in the saliva of healthy subjects, while *Porphyromonas*, *Tannerella*, *Desulfobulbus*, *Eubacterium*, *Phocaeicola* and *Mogibacterium* were associated with the salivary microbiota of periodontitis patients [44]. We can state that these shifts in the saliva composition associated with periodontitis have a direct impact on saliva's protein profiles, which could be used as a signature of the periodontal status. Moreover, the potentiality of saliva in screening patients with periodontitis needs to be investigated further as saliva collection is non-invasive, simple, fast and dispensable of professional skills and training [16].

The decision tree obtained from dental plaque samples showed a sensitivity = 72.1% (± 0.202) and a specificity = 72.2% (± 0.195) for periodontitis diagnosis in blind experiment. We found large variations in peptide profiles (43.5% peaks differentially expressed) in subgingival plaque samples from healthy subjects and periodontitis patients in our study. It was interesting to note that the 10 most differentially expressed peaks in dental plaque had lower levels of intensity in the periodontitis group. The genes *Porphyromonas*, *Treponema*, *Tannerella*, *Filifactor*, and *Aggregatibacter* were found to be more abundant in periodontitis patients, whereas *Streptococcus*, *Haemophilus*, *Capnocytophaga*, *Gemella*, *Campylobacter* and *Granulicatella* were observed at higher levels in healthy subjects [45]. We hypothesize that these peaks could be associated with non-periodontal pathogens as initial colonizers that were less abundant in patients with periodontal diseases. If the average richness of microbes in subgingival plaque from patients with periodontitis was higher than in samples from healthy subjects, the microbial communities associated with periodontal health were correlated with higher Shannon indexes [45]. Subgingival microbiota in periodontal diseases seemed to be characterized by more microbes with low relative abundance compared to healthy microbiota [45]. These findings can explain the lower level of intensity of the top 10 ranking peaks in the periodontitis group.

The concatenation of the decision trees has improved the diagnosis of periodontitis with a sensitivity = 1 (CI 95% = [0.91;1]), a specificity = 1 (CI 95% = [0.92;1]), a positive predictive value = 1 (CI 95% = [0.91;1]) and a negative predictive value = 1 (CI 95% = [0.92;1]). As a consequence, our test provides the best results when at least 2 different samples are tested. Saliva and dental plaque should be preferred because their collection is easier and faster than GCF. In comparison, Tang et al. have evaluated the differentially expressed peaks to distinguish subjects with periodontal diseases from healthy controls with Area Under the Curve's (AUC) values in saliva and in GCF ranged from 0.688 to 0.860 and from 0.926 to 1.000 respectively [30].

To our knowledge, our study was the first to demonstrate that the protein profiles are exclusively specific to periodontal status independently of the periodontal risk factors. These findings make the prospect of a large-scale diagnostic test that could be used without consideration for subjects' conditions other than periodontal status. Previous study exploring

protein profiles in periodontitis diagnostic [30] was conducted in very limited population, free of periodontal risk factors and the results cannot be applied in a large-scale population as periodontitis is strongly linked to several periodontal risk factors [1,46]. By enlarging the selected population and ensuring the independence of the discriminating peaks, we wanted to get closer to the clinical reality.

A large proportion of our periodontitis group was classified into stages III (46,3%) and IV (41,8%), which meant that significant damage to the attachment apparatus has occurred, supporting delays of patients' consultations and professional care. To date, no data related to protein profiles at each stage of periodontitis based upon the new classification (2018) are available. Future studies are needed to investigate whether the early stages (I or II) of periodontitis present the same protein profiles as advanced stages (III or IV).

MALDI-TOF MS is a simple, inexpensive and fast technique that analyses protein profiles with a high reliability rate, and could be used as a rapid screening method in a large population [47]. The observed changes in protein profiles could reflect specific microbiota, as well as the inflammation process with host immune response and periodontal tissue breakdown. The proteomic profile of complex samples like saliva, GCF and dental plaque should be considered as a phenotypic expression resulting from a vast molecular network including bacterial colonization and host immunity. The identification of specific biomarkers responsible for each peak or group of peaks represents a difficult and demanding task that requires further specific studies. The spectra comparison is based on protein fingerprinting. As Maldi-TOF MS is semi-quantitative [20,48], it allows to work on the relative frequency of peaks within the same spectrum, the comparison of these relative frequencies between spectra being ensured by normalization (TIC, total ion current) [20, 49, 50]. This mechanism is the basis of the Maldi-tof bacterial identification technique currently used in routine [20, 51, 52]. In this pilot study, our methodology based on direct sample analysis and profiling obtained good diagnostic performance without identification of specific proteins and has the advantage to not require technical expertise and to be learned by any paramedical personnel. As our strategy was to assess if Microflex LT (Bruker Daltonics, Bremen, Germany) mass spectrometer laser could meet the need of a fast routine diagnostic test in periodontology that could be easily be applied in real conditions, we choose to not freeze the samples compared to the previously-mentioned studies and to carry out the analysis in a short period of time (within 24 to 48 hours). In a study evaluating the right sampling conditions to obtain the best correlation between oral fluid and plasma concentrations of unbound WAR, RR/SS-warfarin and RS/SR-warfarin alcohols, Lomonaco et al. have shown that no significant degradation occurred in the extracted oral fluid samples for at least two months of storage at 4°C [53]. Others substances like cortisol or uric acid remained stable when saliva was stored for at least 4 weeks at 4°C whereas salivary α-amylase activity has been found decreased of about 15% in the same storage conditions [54,55]. For GCF sample, a recent study showed that 10 peaks, over the 20 top signals analyzed, resulted significantly changed after 3 months of storage at -20°C compared to 1 month at -80°C but the authors did not evaluate storage at 4°C and shorter durations [56]. Despite the lack of data available concerning saliva, GCF and dental plaque storage conditions before direct MALDI-TOF MS analysis, the same conditions applied to all of our samples thus ensuring the correct comparability of our results. However, further studies would be necessary to assess the optimal time frame for a routine diagnostic test.

In this pilot study, in order to analyze the performance of the MALDI-TOF test we have randomly chosen a "healthy" population and a periodontitis population according to the criteria of the new classification [1]. We will need to refine the performance with patients from intermediate groups (children, mixed teeth, adolescents, implants, seniors) in further studies.

## Conclusion

In the present study, we developed, for the first time, diagnostic tests based on protein profiles in saliva, GCF and dental plaque between periodontitis patients and healthy subjects. When at least 2 of these samples were tested, the best results were obtained. Because identifying patients at risk of periodontitis remains a challenge in periodontology, it is urgent to develop fast routine biological diagnostic tests, non-invasive and easy to perform by the practitioner.

## Supporting information

**S1 Table. List of epidemiological data recording during the medical questionnaire.**
(PDF)

**S2 Table. List of peaks detected in saliva.**
(PDF)

**S3 Table. List of peaks detected in GCF.**
(PDF)

**S4 Table. List of peaks detected in dental plaque.**
(PDF)

## Acknowledgments

The authors would like to thank Professor Martine Bonnaure-Mallet, Doctors Alexandra Boyer, Elsa Solal, Léa Tholozan, Cathy Dumas, Sébastien Melloul, Mathias Faure-Brac, Camille Sadowski and Basheer Khadaroo for their valuable assistance. English editing of the article was performed by American Journal Expert under ID: Q2D3W36L.

## Author Contributions

**Conceptualization:** Philippe Brouqui, Virginie Monnet-Corti.

**Formal analysis:** Hervé Chaudet.

**Investigation:** Angéline Antezack.

**Software:** Hervé Chaudet.

**Supervision:** Hervé Tissot-Dupont, Philippe Brouqui, Virginie Monnet-Corti.

**Writing – original draft:** Angéline Antezack.

**Writing – review & editing:** Angéline Antezack, Philippe Brouqui, Virginie Monnet-Corti.

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
