## [Decision Letter · Decision Letter 0]

4 Dec 2019

PONE-D-19-31089

Comparative peptide profiles of saliva, gingival crevicular fluid and dental plaque using MALDI-TOF MS on samples obtained from periodontitis patients and healthy control subjects

PLOS ONE

Dear Virgine Monnet-Corti,

Thank you for submitting your manuscript to PLOS ONE. After careful consideration, we feel that it has merit but does not fully meet PLOS ONE’s publication criteria as it currently stands. Therefore, we invite you to submit a revised version of the manuscript that addresses the points raised during the review process.

We would appreciate receiving your revised manuscript by 3th January 2020. To enhance the reproducibility of your results, we recommend that if applicable you deposit your laboratory protocols in protocols.io, where a protocol can be assigned its own identifier (DOI) such that it can be cited independently in the future. For instructions see: http://journals.plos.org/plosone/s/submission-guidelines#loc-laboratory-protocols

We look forward to receiving your revised manuscript.

Kind regards,

Tommaso Lomonaco, Ph.D

Academic Editor

PLOS ONE

Journal Requirements:

Additional Editor Comments:

Dear Authors,

as suggested by the reviewers, the paper requires a major revision before to be accepted in PlosOne journal. In particular all the questions must be addressed properly. I suggest to extend the introduction by discussing the main advantages of saliva compared to other biological fluids (e.g. blood and its derivatives) as well as the possible application of saliva analysis in the field of therapeutic drug monitoring, biomarkers for diseases and etc.

The following articles can be useful for the authors and should be included in the references:

doi.org/10.1371/journal.pone.0028182

doi.org/10.1016/j.microc.2017.02.010

doi.org/10.1007/s00216-019-02158-6

As stated by the reviewer n° 2, the discussion must be implemented considering the fact that saliva collection protocols may alter the chemical composition of sample leading to not reliable data. The following articles can be useful for the authors and used for the discussion:

doi.org/10.1016/j.microc.2017.02.032

doi.org/10.1016/j.microc.2017.04.033

doi.org/10.1371/journal.pone.0114430

Best regards,

Tommaso Lomonaco

Reviewers' comments:

Reviewer's Responses to Questions

**Comments to the Author**

1. Is the manuscript technically sound, and do the data support the conclusions?

Reviewer #1: No

Reviewer #2: No

2. Has the statistical analysis been performed appropriately and rigorously? 

Reviewer #1: No

Reviewer #2: No

3. Have the authors made all data underlying the findings in their manuscript fully available?

Reviewer #1: Yes

Reviewer #2: No

4. Is the manuscript presented in an intelligible fashion and written in standard English?

Reviewer #1: Yes

Reviewer #2: Yes

5. Review Comments to the Author

Reviewer #1: The current article, titled “Comparative peptide profile of saliva, gingival crevicular fluid and dental plaque using MALDI-TOF MS on samples obtained from periodontitis patients ad healthy control subjects”, discusses the differentially expressed protein peaks between two groups of subjects using multiple minimally invasive sample types (saliva, gingival crevicular fluid and dental plaque). Some important questions remain unanswered and require addressing. A major revision is required.

Major comments:

1) How were proteins extracted form each sample? This experimental part in the materials and methods section is lacking. Saliva is an aqueous fluid containing high amount of salts, cells and debris that should be removed in order to investigate the proteomic content. Similarly, is not clear how the proteins were extracted from GCF and subgingival dental plaque samples.

2) From the mass spectrometry paragraph in the materials and methods section is stated that each sample was spotted onto the MALDI target with a volume of 0.5 uL. Was the total protein concentration of each sample evaluated? In order to compare the proteomic profiles of the two groups (Parodontitis and healthy subjects) the amount of proteins from each sample (saliva, GCF and subgingival dental plaque) spotted onto the MALDI target has to be the same. Before mass spectrometry analysis the total protein amount has to be estimated in each sample in order to spot onto the target the same amount of proteins (ug) from each sample. There are different protein assays available on the market to evaluate the total protein concentration.

My concern, is that the differences in the proteomic profiles, observed by the authors, could be indeed affected by the differences in the protein amount of each samples and not related with the disease status of the subjects included in the study.

3) Line 155: “In addition, the usual negative and positive controls were employed for each target”. What is the usual negative and positive control? Please clarify this.

4) Although low resolving power mass spectrometers are used, it should be standard practice to mention external mass spectrometer calibration accuracy.

5) How was the mass spectra pre-processing performed (i.e. mass spectra normalization, peak picking)?

Minor comments:

1) Line 24 – please explain the abbreviation GCF and add: gingival crevicular fluid (GCF).

2) Line 39 – Change “in” with “of”: “peptide profiles in saliva, GCF and dental plaque”.

3) Line 351 – add the word “first”: “In the present study, we developed, for the…time […]”.

4) Lines 394 and 397 – Please remove the “a” and “b” superscript.

5) Please check the order of the Reference List. The first paper cited in the Introduction, Papapanou et al, 2018 correspond to the reference number 20 in the Reference list. Please order the cited papers in order of appearance in the text.

Reviewer #2: The manuscript entitled: “Comparative peptide profiles of saliva, gingival crevicular fluid and dental plaque using MALDI-TOF MS on samples obtained from periodontitis patients and healthy control subjects” aims at determining if peptide profiles of saliva, GCF and dental plaque could be used for discrimination between patients with periodontal diseases and healthy periodontal subjects in a blind test.

Although the issue is interesting, in order to perform comparative peptide profile analysis using MALDI-TOF mass spectrometry the standardization of sample collection and preparation is a fundamental prerequisite. In fact, one of the requirements for a platform to be used in clinical proteomic studies aiming at biomarker discovery is the standardization of the preanalytical and analytical phase, which is essential for the generation of reproducible and robust MS-data. This is completely missing in the present manuscript!

Indeed, the MALDI-TOF MS analysis on samples obtained from 141 patients (67 periodontitis and 74 healthy controls) indicate that this was a mammoth work. However, considering that the standardization of proteomics procedures represents the fundamental prerequisite for diagnostic goals fulfillment, the reviewer recommendation is to reject the work at this stage due to major pitfalls described below in the experimental plan:

• The paper is lacking information about the assays for protein concentration of saliva, GCF and dental plaque and consequently, any kind of normalization for protein concentration has not been performed. The total amount of biological samples was not determined and without this important information it is uncertain whether the statistical difference among disease groups and healthy controls is due to a real difference in protein expression levels or purely reflects different amount of collected samples. These are the fundamental evaluations in order to standardize the protocol and to proceed with MS analysis for comparative studies. The authors should start from this stage to make a quantitative analysis comparable;

• Another important step in the standardization protocol is the storage conditions adopted for the biological samples. The authors wrote that: “Samples were immediately stored at 4°C and analyzed within 24 to 48 hours”. A number of studies demonstrated that the proteins can undergo degradation processes and that collection/handling and storage conditions may influence the stability of endogenous peptidome of biological fluids (Del Boccio et al, Ann Neurol. 2007; del Campo et al., Biomark Med. 2012; Preianò el al., Proteomics. 2016) therefore the authors should have preliminary performed experiments to identify the storage conditions which enable optimal preservation of biological specimen in order to ensure that the variation of protein expression levels in spectra reflects real biological differences rather than experimental artefacts. In this respect, the claims are NOT properly placed in the context of the previous literature.

• The authors should provide more information about MALDI sample preparation protocol, instruments settings adopted for spectra acquisition, and about the parameters for data processing (such as signal to noise and spectra normalization). The authors should investigate the impact of more strictly analytical variables on the generation of reliable MALDI-TOF spectra. It is well known that before starting MALDI based protein profiling study, it is necessary to assess the reliability of profiles with exploratory experiments in order to increase the analytical performances and the robustness of the results; in particular, they should have preliminary analyzed the influence of MALDI sample preparation by modulating the matrix composition and the analyte/matrix ratio in order to optimize the reproducibility of the MALDI peptidome profiles.

• The authors have not shown how reproducible the system is. They have not indicated the number of replicate analyses and the resulting coefficient of variations (CVs) considering that for diagnostic test CVs must fall in a range of 1.5-10% (Albrethsen J. Reproducibility in protein profiling by MALDI-TOF mass spectrometry. Clin Chem 2007;53:852–8).

• The authors must provide more information about the parameters for data processing (such as signal to noise and spectra normalization) and criteria of statistical analysis.

Additionally, it must be considered that for quantitative analysis, it is necessary to use an internal standard, for example an endogenous control protein, that can be observed within the same MS run for all the sample analysis. Alternatively, at least spectra normalization is required.

• The authors wrote: “All the patients were selected based upon periodontal status, regardless of the other criteria (e.g., systemic disease or disease/infection that may affect the periodontal health status, use of antibiotics or immunosuppressant medication within 3 months, current or former smokers), in order to reduce selection bias”. Are the investigators sure that the above mentioned conditions, such as use of antibiotics or anti-inflammatory do not interfere with the analysis of the molecular profile precluding the eligibility of the subjects for the present study?

• Why did the authors decide to consider only the top 10 ranking peaks for the discriminant analysis? I think that this choice could lead to a loss of information.

Finally, some minor points are listed below

• Figure 1 shows spectra obtained from saliva, GCF and dental plaque samples. Do these spectra derive from representative individuals? In order to make data clearer to the reader, the spectra should be shown in the appropriate m/z range for the best detection of molecular features of samples. Moreover, the top 10 ranking peaks found differentially expressed between the periodontitis and the control groups for each type of sample should be highlighted in the figure. Considering that the study focuses on MALDI-MS analysis, the choice of the representative spectra is very important. Are the authors sure that the selected spectra in Fig 1 are the most representative? Particularly, for GCF, the spectra appears poorly resolved and few peaks are characterized by a good S/N ratio.

• Figure 3 should be also revised, in order to make it clearer.

• The figure legends do not have all the information required to easily follow the discussion, so the authors must provide more details to the reader.

• In the discussion, experimental data are not appropriately commented and some link between literature studies and authors findings should be revised. The authors have not treated the literature fairly.

6. PLOS authors have the option to publish the peer review history of their article (what does this mean?). If published, this will include your full peer review and any attached files.

Reviewer #1: No

Reviewer #2: No

---

## [Author Response · Author response to Decision Letter 0]

30 Jan 2020

Response to Reviewer #1: 

We thank you for your review of our manuscript. We have carefully taken into consideration your comments and answered each of your points below. 

Major comments: 

1) How were proteins extracted form each sample? This experimental part in the materials and methods section is lacking. Saliva is an aqueous fluid containing high amount of salts, cells and debris that should be removed in order to investigate the proteomic content. Similarly, is not clear how the proteins were extracted from GCF and subgingival dental plaque samples. 

Answer: We corrected the paragraph “Mass spectrometry” (line 175) in the manuscript in order to better detail our methodology. Proteins were not extracted and samples were directly spotted on steel target and analysed by MALDI-TOS MS. We used the conventional technique described for bacterial identification called profiling (or fingerprinting) which is based on the comparison of spectra of unknown bacteria to database containing spectra of known reference bacteria. In this approach, intact cells can directly be mixed with MALDI matrix (Sandrin TR et al. Mass Spectrom Rev. 2013). Therefore, we chose to use whole samples to generate spectra that were compared to each other in order to assess if MALDI-TOF profiling applied to saliva, gingival crevicular fluid and dental plaque could be used as a diagnostic tool in periodontology. 

2) From the mass spectrometry paragraph in the materials and methods section is stated that each sample was spotted onto the MALDI target with a volume of 0.5 uL. Was the total protein concentration of each sample evaluated? In order to compare the proteomic profiles of the two groups (Parodontitis and healthy subjects) the amount of proteins from each sample (saliva, GCF and subgingival dental plaque) spotted onto the MALDI target has to be the same. Before mass spectrometry analysis the total protein amount has to be estimated in each sample in order to spot onto the target the same amount of proteins (ug) from each sample. There are different protein assays available on the market to evaluate the total protein concentration. My concern, is that the differences in the proteomic profiles, observed by the authors, could be indeed affected by the differences in the protein amount of each samples and not related with the disease status of the subjects included in the study. 

Answer: We used MALDI-TOF MS to perform profiling of saliva, gingival crevicular fluid and dental plaque from periodontitis and healthy subjects, which is not a quantitative analysis. (Sandrin TR et al, Mass Spectrom Rev. 2013). This approach was performed in our laboratory for arthropods identification (El Hamzaoui B et al, PLoS Negl Trop Dis. 2018; Tandina F et al, Parasitology. 2018). 

3) Line 155: “In addition, the usual negative and positive controls were employed for each target”. What is the usual negative and positive control? Please clarify this. 

Answer: We clarified this in the paragraph “Mass spectrometry” (line 175). The calibration of the mass spectrometer is performed with a commercial solution (BTS: Bacterial standard test) (line 190). In addition, in each MALDI target, 4 spots were made with an inactivated strain of Escherichia coli as the positive control (objective score > 2, 1) and 4 spots were made with matrix solution alone as the negative control (objective score <1, 5) (lines 192-194). 

4) Although low resolving power mass spectrometers are used, it should be standard practice to mention external mass spectrometer calibration accuracy. 

Answer: External mass spectrometer calibration accuracy was +/-300ppm. 

5) How was the mass spectra pre-processing performed (i.e. mass spectra normalization, peak picking)? 

Answer: We precised the mass spectra pre-processing in the paragraph “Mass spectrometry” (line 175). Quality criteria of the spectrum for global aspect and intensity were checked: intensity above 104 arbitrary units (AU), horizontal baseline curve and presence of visually identifiable peaks (lines 195-197). 

Minor comments: 

1) Line 24 – please explain the abbreviation GCF and add: gingival crevicular fluid (GCF). 

Answer: We clarified it. 

2) Line 39 – Change “in” with “of”: “peptide profiles in saliva, GCF and dental plaque”. 

Answer: We changed it. 

3) Line 351 – add the word “first”: “In the present study, we developed, for the…time […]”. 

Answer: We added it. 

4) Lines 394 and 397 – Please remove the “a” and “b” superscript. 

Answer: We removed it. 

5) Please check the order of the Reference List. The first paper cited in the Introduction, Papapanou et al, 2018 correspond to the reference number 20 in the Reference list. Please order the cited papers in order of appearance in the text. 

Answer: We modified the order of the reference list. 

Response to Reviewer #2 

We thank you for your review of our manuscript. We totally agreed with all your comments and answered each of your points below. 

Major comments: 

1) The paper is lacking information about the assays for protein concentration of saliva, GCF and dental plaque and consequently, any kind of normalization for protein concentration has not been performed. The total amount of biological samples was not determined and without this important information it is uncertain whether the statistical difference among disease groups and healthy controls is due to a real difference in protein expression levels or purely reflects different amount of collected samples. These are the fundamental evaluations in order to standardize the protocol and to proceed with MS analysis for comparative studies. The authors should start from this stage to make a quantitative analysis comparable. 

Answer: The aim of this pilot study was not to identify biomarker but to assess the feasibility and diagnostic contribution of profiling using MALDI-TOF applied to saliva, GCF and dental plaque from periodontitis and healthy subjects. We clarified it in the introduction of our manuscript. Profiling was described as a conventional approach for bacterial identification by comparing spectra of unknown bacteria to libraries that contain spectra of known reference bacteria. In this technique, intact cells can directly be mixed with MALDI matrix (Sandrin TR et al, Mass Spectrom Rev. 2013). We choose to adapt this non quantitative approach to oral fluids and dental plaque samples to generate spectra that were compared to each other and to evaluate if they could classify subjects according to their periodontal status. This approach has already been performed in our laboratory for arthropods identification (El Hamzaoui B et al, PLoS Negl Trop Dis. 2018; Tandina F et al, Parasitology. 2018). 

2) Another important step in the standardization protocol is the storage conditions adopted for the biological samples. The authors wrote that: “Samples were immediately stored at 4°C and analyzed within 24 to 48 hours”. A number of studies demonstrated that the proteins can undergo degradation processes and that collection/handling and storage conditions may influence the stability of endogenous peptidome of biological fluids (Del Boccio et al, Ann Neurol. 2007; del Campo et al., Biomark Med. 2012; Preianò el al., Proteomics. 2016) therefore the authors should have preliminary performed experiments to identify the storage conditions which enable optimal preservation of biological specimen in order to ensure that the variation of protein expression levels in spectra reflects real biological differences rather than experimental artefacts. In this respect, the claims are NOT properly placed in the context of the previous literature. 

Answer: We thank you for this pertinent remark and we argued our choice of methodology in the discussion. Del Boccio et al. (2007) and Del Campo et al. (2012) analysed cerebrospinal fluid among different storage conditions. As we did not use the same samples, we couldn’t use the results of these two studies, even though Del Campo et al. (2012) “recommended to store the samples at 4°C, since it can be easily done and it does not modify the biochemical results”. Preianò et al. (2016) found that best preserved signatures were obtained when GCF samples were stored at -80°C for 1 month compared to samples stored at -20°C during 3 months but the authors did not evaluate storage at 4°C and shorter durations. As Bellagambi et al. (Microchem J, 2018) and Lomonaco et al. (PLoS One, 2014; Microchem. J, 2018) found, oral fluids storage at 4°C during 4 weeks can preserve substances like cortisol, uric acid or WAR, RR/SS-warfarin and RS/SR-warfarin alcohols. Despite the lack of data available concerning saliva, GCF and dental plaque storage conditions before direct MALDITOF MS analysis, the same conditions applied to all of our samples thus ensuring the correct comparability of our results. 

3) The authors should provide more information about MALDI sample preparation protocol, instruments settings adopted for spectra acquisition, and about the parameters for data processing (such as signal to noise and spectra normalization). The authors should investigate the impact of more strictly analytical variables on the generation of reliable MALDI-TOF spectra. It is well known that before starting MALDI based protein profiling study, it is necessary to assess the reliability of profiles with exploratory experiments in order to increase the analytical performances and the robustness of the results; in particular, they should have preliminary analyzed the influence of MALDI sample preparation by modulating the matrix composition and the analyte/matrix ratio in order to optimize the reproducibility of the MALDI peptidome profiles. 

Answer: Our study was a pilot study and we clarified it in the introduction of our manuscript. The aim was to assess the feasibility and diagnostic contribution of protein profiling using MALDI-TOF applied to saliva, GCF and dental plaque from periodontitis and healthy subjects. We hypothesized that rapid routine and blinded MALDI-TOF analysis could accurately classify these three types of samples according to the periodontal state. We chose a methodology following as closely as possible the conventional MALDI-TOF protocol (used in routine diagnostic, i.e Microflex LT, Brucker and a mass range of 2-20 kDa) and we did not look for biomarkers identification. 

4) The authors have not shown how reproducible the system is. They have not indicated the number of replicate analyses and the resulting coefficient of variations (CVs) considering that for diagnostic test CVs must fall in a range of 1.5-10% (Albrethsen J. Reproducibility in protein profiling by MALDI-TOF mass spectrometry. Clin Chem 2007; 53:852–8). 

Answer: The number of replicate analyses was 6 (line 176 and line 184). The spectra generated for each sample of each subject were merged in order to develop a main spectra (MSP) (lines 202-205). 

5) The authors must provide more information about the parameters for data processing (such as signal to noise and spectra normalization) and criteria of statistical analysis. Additionally, it must be considered that for quantitative analysis, it is necessary to use an internal standard, for example an endogenous control protein that can be observed within the same MS run for all the sample analysis. Alternatively, at least spectra normalization is required. 

Answer: We provided more informations about the parameters for date processing in paragraphs “Mass spectrometry” (line 175) and “Bioinformatics analysis” (line 199). The signal-to-noise applied was = 3 (line 204). The calibration of the MS was fully automated and performed with a commercial solution (BTS: Bacterial standard test) and the procedure was completely automatic (BiotyperRTC user manual) (lines 189192). 

6) The authors wrote: “All the patients were selected based upon periodontal status, regardless of the other criteria (e.g., systemic disease or disease/infection that may affect the periodontal health status, use of antibiotics or immunosuppressant medication within 3 months, current or former smokers), in order to reduce selection bias”. Are the investigators sure that the above mentioned conditions, such as use of antibiotics or anti-inflammatory do not interfere with the analysis of the molecular profile precluding the eligibility of the subjects for the present study? 

Answer: The aim of our pilot study was to assess the feasibility and diagnostic contribution of profiling using MALDI-TOF applied to saliva, GCF and dental plaque from periodontitis and healthy subjects. This idea has already been investigated by Tand et al. (Clin Chim Acta. 2019) using saliva, GCF and serum samples in a very limited population, free of periodontal risk factors. In such conditions, their results deviated from target population and could not be applied to a routine large-scale diagnostic test. That’s why we decided to randomly choose a large healthy and periodontitis population in order to be closer to the clinical reality and to assess whether there were co-factors that could impact on the diagnosis. 

7) Why did the authors decide to consider only the top 10 ranking peaks for the discriminant analysis? I think that this choice could lead to a loss of information. 

Answer: We have chosen the 10 most discriminating peaks because only 7 were needed to build the diagnostic decision tree for saliva and only 9 for GCF and dental plaque. The objective was to select the minimum number of peaks to obtain the best sensitivity and specificity (lines 271-272). 

Minor comments: 

• Figure 1 shows spectra obtained from saliva, GCF and dental plaque samples. Do these spectra derive from representative individuals? In order to make data clearer to the reader, the spectra should be shown in the appropriate m/z range for the best detection of molecular features of samples. Moreover, the top 10 ranking peaks found differentially expressed between the periodontitis and the control groups for each type of sample should be highlighted in the figure. Considering that the study focuses on MALDI-MS analysis, the choice of the representative spectra is very important. Are the authors sure that the selected spectra in Fig 1 are the most representative? Particularly, for GCF, the spectra appears poorly resolved and few peaks are characterized by a good S/N ratio. 

Answer : You are right, we have chosen an example that we are not sure is representative. We removed the figure. 

• Figure 3 should be also revised, in order to make it clearer. 

Answer : We totally agree with your remark and we removed the figure. 

• The figure legends do not have all the information required to easily follow the discussion, so the authors must provide more details to the reader. 

Answer : We removed figure 2 and figure 3 that were not clear. 

• In the discussion, experimental data are not appropriately commented and some link between literature studies and authors findings should be revised. The authors have not treated the literature fairly. 

 Answer : You are right and we clarified our manuscript. 

Response to Academic Editor, Tommaso Lomonaco, Ph.D 

We thank you for considering our manuscript and for allowing us the opportunity to respond to the peer review comments. We found all the comments very helpful to improve our manuscript. We thank you for the six articles that you suggested and we added them, as you advised us, in the introduction and in the discussion. 

We hope now that the revised manuscript will be suitable for publication.

---

## [Decision Letter · Decision Letter 1]

19 Feb 2020

PONE-D-19-31089R1

Rapid diagnosis of periodontitis, a feasibility study using MALDI-TOF mass spectrometry

PLOS ONE

Dear Virginie Monnet-Corti,

Thank you for submitting your manuscript to PLOS ONE. After careful consideration, we feel that it has merit but does not fully meet PLOS ONE’s publication criteria as it currently stands. Therefore, we invite you to submit a revised version of the manuscript that addresses the points raised during the review process.

We would appreciate receiving your revised manuscript by 25 February. To enhance the reproducibility of your results, we recommend that if applicable you deposit your laboratory protocols in protocols.io, where a protocol can be assigned its own identifier (DOI) such that it can be cited independently in the future. For instructions see: http://journals.plos.org/plosone/s/submission-guidelines#loc-laboratory-protocols

We look forward to receiving your revised manuscript.

Kind regards,

Tommaso Lomonaco, Ph.D

Academic Editor

PLOS ONE

Additional Editor Comments (if provided):

Dear Authors,

the paper requires additional revisions before to be accepted in Plos One. Generally, I suggest to emphasize that MALDI-TOF approach was used to preliminary investigate the composition of saliva collected from patients suffering from periodontitis. Regarding the title,

I prefer the new version.

Please clarify the following aspects:

1. I suggest to move the table 1 in the supplementary information.

2. Please explain at which variable the p-value was related (table 2).

3. please explain the meaning of the percentage in the brackets (table 3).

4. please explain the meaning of the number in the brackets (L374).

5. please include the RSD regarding replicate analysis on the same sample.

Regards,

Tommaso Lomonaco

Reviewers' comments:

Reviewer's Responses to Questions

**Comments to the Author**

1. If the authors have adequately addressed your comments raised in a previous round of review and you feel that this manuscript is now acceptable for publication, you may indicate that here to bypass the “Comments to the Author” section, enter your conflict of interest statement in the “Confidential to Editor” section, and submit your "Accept" recommendation.

Reviewer #1: (No Response)

Reviewer #2: (No Response)

2. Is the manuscript technically sound, and do the data support the conclusions?

Reviewer #1: No

Reviewer #2: Partly

3. Has the statistical analysis been performed appropriately and rigorously? 

Reviewer #1: No

Reviewer #2: I Don't Know

4. Have the authors made all data underlying the findings in their manuscript fully available?

Reviewer #1: No

Reviewer #2: Yes

5. Is the manuscript presented in an intelligible fashion and written in standard English?

Reviewer #1: Yes

Reviewer #2: Yes

6. Review Comments to the Author

Reviewer #1: The paper was not properly improved and I can not suggest its publication. The experimental design is not technically sound. In order to determine differentially expressed proteins (as stated even in the abstract, lines 33-34) he authors should have determined the total protein concentration of each sample and to spot the same amount of proteins, as a consequence all of the results are not consistent.

Reviewer #2: In the revised version of the manuscript PONE-D-19-31089R1 entitled: “Rapid diagnosis of periodontitis, a feasibility study using MALDI-TOF mass spectrometry” the Authors have improved the overall quality of the study however the manuscript still remains scientifically and methodologically not accurate because of

1) lack of experiments concerning the short-term storage at 4° of saliva, GCF and dental plaque. For comparative profiling studies, storage time must be the same for all the samples analyzed: the protein profile of a sample stored for 48 h at 4° may not be compared to that stored for 24 h at 4°. Storage time must be standardized to ensure the comparability of the results.

2) lack of precise quantification (for example the lack in the use of internal standards for MALDI-TOF analysis; moreover none of the various kinds of normalization for protein concentration has been performed), especially for peptide and protein profile obtained by MALDI-TOF MS, might be very dangerous because MALDI-TOF is not inherently quantitative.

These limitations could affect the results of the statistical analysis and raise doubts whether the statistical difference among disease groups and healthy controls is due to a real difference in protein expression levels or purely reflects different amounts of collected samples.

Therefore this reviewer strongly suggests that the limitations of this study should be highlighted and openly and critically discussed by the Authors.

All these issues are much more imperative in a “feasibility” study as the Authors claim in the new proposed title of the revised manuscript “Rapid diagnosis of periodontitis, a feasibility study using MALDI-TOF mass spectrometry”

It would be more correct, for the benefit of the readers, to replace the new proposed title “Rapid diagnosis of periodontitis, a feasibility study using MALDI-TOF mass spectrometry”

with a new one for example “Rapid diagnosis of periodontitis, a feasibility study using routine and blinded MALDI-TOF analysis”. In the opinion of this reviewer it is better to clarify the non-conventional use of MALDI-TOF mass spectrometry in this study starting from the title.

Considering that the data presented here are very weak because not well supported by the requested experiments, systematic bias in the design of this study may cause erroneous results causing false expectations in the wide readership of PLos ONE. Nevertheless, in a reasonable scenario it could be also possible that such study could lead to controversial debate on a possible misuse of MALDI as already happened for SELDI pioneering studies.

7. PLOS authors have the option to publish the peer review history of their article (what does this mean?). If published, this will include your full peer review and any attached files.

Reviewer #1: No

Reviewer #2: No

---

## [Author Response · Author response to Decision Letter 1]

25 Feb 2020

Dear Editor,

We would like to thank you and the Reviewers for your comments on our manuscript. We have edited our manuscript and our point-by-point responses to the comments are outlined below. 

Sincerely,

Professor MONNET-CORTI Virginie, DDS-Ph.D

---

## [Editor Report · Decision Letter 2]

27 Feb 2020

Rapid diagnosis of periodontitis, a feasibility study using MALDI-TOF mass spectrometry

PONE-D-19-31089R2

Dear Dr. Virginie Monnet-Corti,

We are pleased to inform you that your manuscript has been judged scientifically suitable for publication and will be formally accepted for publication once it complies with all outstanding technical requirements.

With kind regards,

Tommaso Lomonaco, Ph.D

Academic Editor

PLOS ONE

Additional Editor Comments (optional):

Dear Authors, all the points were discussed in the revised versione of the paper and thus the manuscript can be published in PlosOne.

Regards,

Tommaso Lomonaco

---

## [Editor Report · Acceptance letter]

2 Mar 2020

PONE-D-19-31089R2 

Rapid diagnosis of periodontitis, a feasibility study using MALDI-TOF mass spectrometry 

Dear Dr. Monnet-Corti:

I am pleased to inform you that your manuscript has been deemed suitable for publication in PLOS ONE. Congratulations! Your manuscript is now with our production department. 

With kind regards,

on behalf of

Dr. Tommaso Lomonaco 

Academic Editor

PLOS ONE